# A prospective study to explore the relationship between *MTHFR C677T* genotype, physiological folate levels, and postpartum psychopathology in at-risk women

Emily Morris[1,2], Catriona Hippman[2], Arianne Albert[3], Caitlin Slomp[2], Angela Inglis[1,2], Prescilla Carrion[2], Rolan Batallones[2], Heather Andrighetti[2], Colin Ross[4,5], Roger Dyer[6], William Honer[2], Jehannine Austin[1,2]*

1 Department of Medical Genetics, University of British Columbia, Vancouver, BC, Canada, 2 Department of Psychiatry, University of British Columbia, Vancouver, BC, Canada, 3 Women's Health Research Institute, Vancouver, BC, Canada, 4 Faculty of Pharmaceutical Sciences, University of British Columbia, Vancouver, BC, Canada, 5 BC Children's Hospital Research Institute, Vancouver, BC, Canada, 6 Analytical Core for Metabolomics & Nutrition (ACMaN), BC Children's Hospital Research Institute, Vancouver, BC, Canada

* jehannine.austin@ubc.ca

**Data Availability Statement:** Our data contains potentially sensitive information. Further, in order to share de-identified data, our Research Ethics

## Abstract

### Background

The etiology of postpartum psychopathologies are not well understood, but folate metabolism pathways are of potential interest. Demands for folate increase dramatically during pregnancy, low folate level has been associated with psychiatric disorders, and supplementation may improve symptomatology. The *MTHFR C677T* variant influences folate metabolism and has been implicated in depression during pregnancy.

### Objective

To conduct a prospective longitudinal study to explore the relationship between *MTHFR C677T* genotype, folate levels, and postpartum psychopathology in at-risk women.

### Hypothesis

In the first three months postpartum, folate will moderate a relationship between *MTHFR* genotype and depression, with *TT* homozygous women having more symptoms than *CC* homozygous women.

### Methods

We recruited 365 pregnant women with a history of mood or psychotic disorder, and at 3 postpartum timepoints, administered the Edinburgh Postnatal Depression Scale (EPDS); Clinician-Administered Rating Scale for Mania (CARS-M) and the Positive and Negative Symptom Scale (PANSS) and drew blood for genotype/folate level analysis. We used robust linear regression to investigate interactions between genotype and folate level on the

Board requires that participants explicitly consent. Unfortunately this was not included in our consent form for this study and we are unable to provide a de-identified data set to be made accessible. Data access requests can be sent to the University of British Columbia Clinical Research Ethics board (ethics.research.ubc.ca).

**Funding:** This work was made possible through funding from the Canadian Institutes of Health Research (CIHR), JA was supported by BC Mental Health and Substance Use Services, the Canada Research Chairs Program, the Michael Smith Foundation for Health Research, and CIHR. JA received funding from the Canadian Institutes of Health Research (https://cihrirsc.gc.ca/e/193.html) to conduct this work (CIHR MOP 82750). The funders had no role in study design, data collection and analysis, decision to publish, or preparation of the manuscript.

**Competing interests:** The authors have declared that no competing interests exist.

highest EPDS and CARS-M scores, and logistic regression to explore interactions with PANSS psychosis scores above/below cut-off.

## Results

There was no significant interaction effect between *MTHFR* genotype and folate level on highest EPDS (p = 0.36), but there was a significant interaction between genotype, folate level and log(CARS-M) (p = 0.02); post-hoc analyses revealed differences in the effect of folate level between *CC/CT*, and *TT* genotypes, with folate level in *CC* and *CT* having an inverse relationship with symptoms of mania, while there was no relationship in participants with *TT* genotype. There was no significant interaction between *MTHFR* genotype and folate level on the likelihood of meeting positive symptom criteria for psychosis on the PANSS (p = 0.86).

## Discussion

These data suggest that perhaps there is a relationship between *MTHFR C677T*, folate level and some symptoms of postpartum psychopathology.

## Introduction

Postpartum psychiatric disorders are urgent health concerns that have important implications for mothers, infants, and their families. Although all women are at risk for an episode of mental illness in the postpartum [1, 2], those who have a history of a mood or psychotic disorder are at greater risk compared to the general population [2–10].

Similar to non-perinatal psychiatric disorders, postpartum psychiatric disorders are thought to arise due to the combined effects of genetic and environmental factors. There is an abundance of literature investigating the role of genetic variations in psychiatric disorders such as schizophrenia and bipolar disorder [11], with accumulating research on gene-environment interactions [12]. However, in the context of postpartum psychiatric disorders, the majority of investigations focus on either environmental contributors or the role of genetics [13–18] with few studies investigating gene and environment interactions (e.g. 5HTTLPR/ monoaminergic variations and stress [19, 20]).The need for studies of postpartum depression that integrate these elements has been recognized [21], and there are potential gene-environment interactions worthy of investigation in relation to postpartum psychopathology. One such potential example involves variations in the gene encoding the enzyme methylenetetrahydrofolate reductase, and folate.

Methyltetrahydrofolate reductase (MTHFR) is a folate dependent enzyme that has a common, functional, thermolabile variant—*MTHFR C677T*. It has been studied in the context of non pregnancy related psychiatric disorders, depression, bipolar disorder, and schizophrenia [20–22]–including a study of the relationship between psychopathology, folate, and the *MTHFR C677T* variant [22].

*MTHFR C677T* variants and low folate levels have also been *separately* studied, and associated with depressed mood *during* pregnancy [23, 24]. Folate deficiency has also been postulated as a contributor to postpartum psychosis and depression [24, 25], but no studies, to our knowledge, have explored whether the *MTHFR C677T* variant increases risk for *postpartum* psychiatric disorders and/or explored how postpartum physiological folate levels interact with

these genotypes in relation to risk. Further, no studies to our knowledge have investigated the role of *MTHFR C677T* or folate levels in postpartum mania. Given the increased demands for folate during pregnancy, and the impact of folate deficiency on *TT* homozygous women [26], there is need to further understand the role of *MTHFR C677T* and folate levels on postpartum psychopathology, especially since there has been some suggestion of using folic acid or L-methylfolate supplements to treat low mood in the postpartum and the general population in those with *MTHFR C677T* variants [27, 28]. A more thorough understanding of genetic risk factors that may suggest remediatory biological interventions (e.g. perinatal folic acid supplementation tailored to *MTHFR* genotype) is critical to improving outcomes for women at risk for postpartum psychopathology.

The purpose of this study was to conduct a prospective, longitudinal, observational study to better understand the relationships between the *MTHFR C677T* variant, physiological folate levels and postpartum depression in at-risk women. We aimed to test the hypothesis that in the first three months postpartum, compared to *MTHFR CC* homozygous women, *TT* homozygous women would have increased symptoms of postpartum depression (PPD) and that this relationship would be moderated by physiological levels of red blood cell (RBC) folate. We also conducted exploratory analyses regarding the impact of *MTHFR C677T* genotypes and RBC folate levels on: a) postpartum mania (PPM) and b) postpartum psychosis (PPP).

## Materials and methods

The study was approved by the University of British Columbia Research Ethics Board (H06-70145). All participants provided written informed consent.

Participants were recruited from the metropolitan Vancouver, Canada area between 2007 and 2016 ($N$ = 365). Women were eligible to participate if they: a) had a history of a mood or psychotic disorder (depression, bipolar disorder, schizophrenia) as confirmed by the Structured Clinical Interview for Diagnosis (SCID) [29]; b) were pregnant; and c) were fluent in English. Details regarding recruitment methods are described elsewhere [30].

The study was observational; no experimental interventions were provided to participants. Data collection occurred at 4 timepoints: T1 (during pregnancy (>15 weeks gestation)); T2 (1–2 week(s) postpartum); T3 (1–2 month(s) postpartum); and T4 (3–4 months postpartum).

At T1, we collected demographic information, and at each timepoint, blood was drawn (for measuring RBC folate, and *MTHFR* genotyping). Also at each timepoint participants completed the Edinburgh Postnatal Depression Scale (EPDS) to assess depression symptomatology (see below), and a trained researcher administered the Clinician Administered Rating Scale (CARS-M) to assess mania symptomatology (see below), and the Positive and Negative Symptom Scale (PANSS) to assess psychosis symptomatology (see below). Participants also provided information regarding use of folic acid supplements and psychotropic medication at each timepoint.

Data were managed using REDCap (Research Electronic Data Capture) tools hosted at BC Children's Hospital Research Institute. REDCap is a secure, web-based application designed to support data capture for research studies [31].

### Instruments

**Edinburgh Postnatal Depression Scale (EPDS).** The EPDS is a 10-item, self-administered, Likert scale-based questionnaire (each item is rated by selecting from 4 options, scored from 0 to 3) that has been validated for measuring both prenatal and postpartum depression [32]. Higher scores indicate greater depression symptomatology.

**Clinician Administered Rating Scale for Mania (CARS-M).** The 10-item CARS-M is a clinician rated, reliable and valid measure of the severity of mania symptomatology with or without psychotic features. On the basis of an interview and observation, severity of symptomatology is assessed by rating each item from 0 (absent) to 5 (extreme) [33].

**Positive and Negative Syndrome Scale (PANSS).** The PANSS is a well-validated instrument (completed by a clinically trained rater, on the basis of a 30–45 minute semi-structured interview) that measures the presence and severity of 30 psychiatric symptoms [34]. Each symptom is rated on a 7-point scale; a score of 1 means the symptom is not present, and a score of 7 means that it is present to an extreme degree. Five of the PANSS items can be used to assess the presence and severity of psychosis, using specific cut off scores [35, 36]. Specifically we categorized psychosis as present if a participant met at least one of the following threshold scores (delusions ≥3, conceptual disorganization ≥5 [37], hallucinations ≥3, suspiciousness ≥5, unusual thought content ≥4).

To ensure agreement between multiple raters (inter-rater concordance) for the PANSS, we conducted periodic PANSS inter-rater concordance sessions and calculated coefficient of agreement (determined by % agreement within 1 rating point amongst all raters) for each of the five PANSS items for psychosis [38].

## Biological measures

**Folate.** RBC folate was measured using an Abbott Architect i1000 immunoanalyzer (Abbott Diagnostics,Canada, Mississauga Ont), folate reagent kit (#B1P740) and Abbott folate calibrators and quality controls. RBC folate levels were corrected for hematocrit and plasma folate.

*MTHFR C677T* **genotyping.** DNA was extracted from buffy coats from whole blood (or if blood samples were unavailable, buccal swabs) using Qiagen QIAmp DNA cultured cells protocol. Real-time PCR using Taqman primers/probes determined genotypes of *MTHFR* C677T (ie *CC*, *CT*, *or TT*).

## Analysis

To allow for differences in timing for the emergence of symptoms of depression and mania we selected the highest postpartum EPDS and CARS-M scores (from all available postpartum timepoints) for each participant and the corresponding RBC folate levels (i.e. RBC folate measured at the time of the highest EPDS or CARS-M) for analysis. When the highest score persisted for more than one timepoint, we used the earliest timepoint with available corresponding RBC folate level. Similarly, to allow for differences in timing for the emergence of key positive symptoms of psychosis, we used the earliest postpartum timepoint for which the participant met criteria for psychosis (determined by the PANSS scores defined above) with available corresponding RBC folate data. If the participant never met criteria for psychosis we used the earliest timepoint with available RBC folate data.

To investigate relationships between: depression (EPDS) or mania (CARS-M) symptoms in the postpartum, MTHFR genotype, and RBC folate levels, we used robust linear regression with MM estimation to mitigate the effects of outliers and highly influential points [39, 40] as implemented in the 'robustbase' package [41]. The presence of outliers and influential data was assessed using diagnostic regression plots of non-robust models. Additionally, the results of the robust regressions were different enough from the non-robust regressions to justify their use. Moderation of the relationship between MTHFR and EPDS or CARS-M by RBC folate levels, was tested by including an interaction term in the models [42]. If a significant interaction term was detected, we estimated the difference in slope among the CC, CT, and TT

genotypes using asymptotic Chi-square tests [43] with Benjamini-Hochberg false-discovery rate p-value correction [44]. Otherwise, the interaction term was removed and the main effects of MTHFR genotype and RBC folate were estimated. CARS-M scores were log transformed to better meet the assumptions of normality and homogeneity of variance. We calculated Cohen's d values to estimate the effect size of the difference in mean EPDS and CARS-M scores between CC, CT, and TT genotypes, controlled for RBC folate.

We investigated relationships between: likelihood of meeting criteria for psychosis on the PANSS, *MTHFR* genotype, and RBC folate levels with logistic regression.

Statistical significance was assumed at $p < 0.05$.

We used descriptive statistics to report demographic data. To test differences in demographic and clinical variables between genotypes we used Kruskal-Wallis tests for continuous variables and Fisher Exact tests for categorical variables.

## Results

The study enrolled 365 women. Sufficient postpartum data was collected from 327 participants to be included in analyses (i.e. completed the PANSS, EPDS, or CARS-M, had corresponding RBC folate data for at least one postpartum timepoint, and were genotyped for *MTHFR C677T*). There were no significant differences in demographic characteristics between *MTHFR* genotype groups, and distribution of genotype groups did not deviate from Hardy-Weinberg equilibrium ($\chi^2 = 3.67$, df = 1, p = 0.06) (see Table 1).

### Depression

There were 305 participants whose highest postpartum EPDS had a corresponding RBC folate available (see Table 2).

RBC folate and highest postpartum EPDS scores (EPDS vs RBC folate) are shown for each *MTHFR* genotype in Fig 1. There was no significant interaction between *MTHFR* genotype and RBC folate level on highest EPDS (p = 0.36). There was also no relationship between RBC folate and EPDS on its own (coefficient = -0.002 (95%CI = -0.004 to 0.0008), p = 0.19, adjusted R-squared = 0.002), and no difference between genotypes for mean EPDS controlling for RBC folate (p = 0.59, Cohen's d values: *CC* compared to *CT*: d = 0.02, *CC* compared to *TT*: d = 0.16, and *CT* compared to *TT*: d = 0.18).

### Mania

There were 233 women whose highest CARS-M had a corresponding RBC folate level (Table 3).

RBC folate level and log transformed CARS-M (CARS-M vs RBC folate) scores are shown for each *MTHFR* genotype in Fig 2. There was a significant interaction between *MTHFR* genotype and RBC folate level on highest (log)CARS-M scores (p = 0.02). Post-hoc analyses suggested no significant differences between the slopes (CARS-M vs RBC Folate) for *MTHFR CC* and *CT* genotypes (p = 0.65), but suggested that there was a significant difference between the slopes of CARS-M vs RBC folate for *MTHFR CC* and *TT* (p = 0.03) genotypes, and between the slopes of CARS-M vs RBC folate for *MTHFR CT* and *TT* (p = 0.03) genotypes (Fig 2). The slopes for the *CC* and *CT* genotype groups showed an inverse relationship between RBC folate and CARS-M, with higher folate levels associated with lower CARS-M scores. However, the slope for the *TT* genotype suggests that there was not a strong relationship between CARS-M and RBC folate within the *TT* genotype group.

**Table 1. Demographics for each *MTHFR C677T* genotype.**

| | Total | MTHFR genotype | | | P-value |
| --- | --- | --- | --- | --- | --- |
| | | TT | CT | CC | |
| | N = 327 | n = 37 | n = 125 | n = 165 | |
| **Age (years) (n = 326)** | | | | | |
| Mean (SD) | 31.1 (±5.7) | 31.0 (±6.4) | 30.8 (±5.9) | 31.3 (±5.4) | 0.83 |
| **Annual Household Income ($ CAD) (n = 316)** | | | | | |
| <$20,000 | 32 (10.1%) | 4 (11.1%) | 12 (9.8%) | 16 (10.1%) | 0.16 |
| $20,000 - $40,000 | 54 (17.1%) | 11 (30.6%) | 24 (19.7%) | 19 (12.0%) | |
| $41,000 - $60,000 | 55 (17.4%) | 6 (16.7%) | 21 (17.2%) | 28 (17.7%) | |
| $61,000 - $80,000 | 62 (19.6%) | 6 (16.7%) | 24 (19.7%) | 32 (20.3%) | |
| $81,000 - $100,000 | 44 (13.9%) | 6 (16.7%) | 19 (15.6%) | 19 (12.0%) | |
| >$100,000 | 69 (21.8%) | 3 (8.3%) | 22 (18.0%) | 44 (27.8%) | |
| **Ethnicity (n = 321)** | | | | | |
| European | 243 (75.7%) | 24 (66.7%) | 91 (72.8%) | 128 (80.0%) | 0.38 |
| Asian | 24 (7.5%) | 3 (8.3%) | 8 (6.4%) | 13 (8.1%) | |
| Mixed | 42 (13.1) | 7 (19.4%) | 20 (16.0%) | 15 (9.4%) | |
| Other* (includes African and Aboriginal) | 10 (3.1%) | 2 (5.6%) | 5 (4.0%) | 3 (1.9%) | |
| **Highest level of education (n = 315)** | | | | | |
| High school | 24 (7.6%) | 4 (11.4%) | 10 (8.3%) | 10 (6.3%) | 0.79 |
| Less than 4 years of college/university | 121 (38.4%) | 10 (28.6%) | 47 (38.8%) | 64 (40.3%) | |
| 4 or more years of college/university | 157 (49.8%) | 20 (57.1%) | 58 (47.9%) | 79 (49.7%) | |
| Did not complete high school | 13 (4.1%) | 1 (2.9%) | 6 (5.0%) | 6 (3.8%) | |
| **Employment (n = 326)** | | | | | |
| Not employed | 48 (14.7%) | 4 (10.8%) | 23 (18.4%) | 21 (12.8%) | 0.34 |
| Employed | 278 (85.3%) | 33 (89.2%) | 102 (81.6%) | 143 (87.2%) | |
| **Body Mass Index (BMI) (n = 296)** | | | | | |
| Mean (SD) | 29.1 (±5.6) | 28.7 (±5.4) | 28.7 (±5.5) | 29.4 (±5.7) | 0.74 |
| **Gravida (n = 324)** | | | | | |
| Median (IQR) | 2.0 (1.0–3.0) | 2.0 (1.0–3.0) | 2.0 (1.0–3.0) | 2.0 (1.0–3.0) | 0.79 |
| **Number of children at enrollment (n = 324)** | | | | | |
| Median (IQR) | 0.0 (0.0–1.0) | 0.0 (0.0–1.0) | 0.0 (0.0–1.0) | 0.0 (0.0–1.0) | 0.83 |
| **IVF pregnancy (n = 320)** | | | | | |
| no | 307 (95.9%) | 33 (94.3%) | 115 (94.3%) | 159 (97.5%) | 0.26 |
| yes | 13 (4.1%) | 2 (5.7%) | 7 (5.7%) | 4 (2.5%) | |
| **Marital status (n = 324)** | | | | | |
| Married/Common-Law/Partnered | 302 (93.2%) | 33 (97.3%) | 109 (88.6%) | 157 (95.7%) | 0.04 |
| Single | 22 (6.8%) | 1 (2.7%) | 14 (11.4%) | 7 (4.3%) | |
| **Psychiatric Diagnosis (n = 326)** | | | | | |
| Bipolar Disorder | 60 (18.4%) | 7 (18.9%) | 21 (16.8%) | 32 (19.5%) | 0.95 |
| Depression | 264 (81.3%) | 30 (81.1%) | 103 (82.4%) | 131 (79.9%) | |
| Schizophrenia | 2 (0.6%) | 0 (0.0%) | 1 (0.8%) | 1 (0.6%) | |
| **Previous History of psychotic symptoms (n = 326)** | | | | | |
| no | 251 (77.0%) | 28 (75.7%) | 102 (81.6%) | 121 (73.8%) | 0.28 |
| yes | 75 (23.0%) | 9 (24.3%) | 23 (18.4%) | 43 (26.2%) | |
| **Psychotropic medication in the postpartum (n = 325)** | | | | | |
| no | 216 (66.5%) | 24 (64.9%) | 84 (67.7%) | 108 (65.9%) | 0.93 |
| yes | 109 (33.5%) | 13 (35.1%) | 40 (32.3%) | 56 (34.1%) | |
| **RBC folate (ng/ml)** | | | | | |

*(Continued)*

**Table 1.** (Continued)

| | | MTHFR genotype | | | |
| --- | --- | --- | --- | --- | --- |
| | Total | TT | CT | CC | P-value |
| | N = 327 | n = 37 | n = 125 | n = 165 | |
| Mean (SD) | 663.5 (261.7) | 777.0 (332.3) | 650.0 (198.4) | 648.2 (189.8) | 0.097 |
| **Taking folic acid supplement throughout postpartum (n = 326)** | | | | | |
| no | 127 (39.0%) | 12 (33.3%) | 48 (38.4%) | 67 (40.6%) | 0.71 |
| yes | 199 (61.0%) | 24 (66.7%) | 77 (61.6%) | 98 (59.4%) | |

## Psychosis

Coefficient of agreement for each of the five PANSS items (delusions, conceptual disorganization, hallucinations, suspiciousness, and unusual thought content) from PANSS interrater concordance sessions held over the course of the study were 0.85, 0.70, 0.90, 0.77, and 0.85 with an overall mean coefficient of agreement of 0.82.

There were 326 women with sufficient postpartum PANSS data and corresponding RBC folate data, for at least one postpartum timepoint (Table 4).

There was no significant interaction between *MTHFR* genotype and RBC folate level on the probability of meeting psychotic symptom criteria on the PANSS (p = 0.86). RBC folate levels and whether participants met criteria for psychosis for each *MTHFR* genotype is shown in Fig 3. There was also no relationship between RBC folate and psychotic symptoms on its own (OR = 1.00, 95%CI = 0.99 to 1.01, p = 0.09), and there was also no difference between genotypes for psychotic symptoms controlling for RBC folate (p = 0.86; Odds Ratio (OR) for *CT* compared to *CC* = 1.16 (95%CI = 0.66 to 1.03); OR for *TT* compared to *CC* = 1.12 (95%CI = 0.44 to 1.60)).

Regression coefficients and data from both linear (depression and mania) and logistic (psychosis) regression analyses are displayed in S1 Table.

## Discussion

This is the first study to our knowledge to explore relationships between *MTHFR C677T* variants, physiological levels of folate, and postpartum psychiatric symptoms, and specifically it is

**Table 2. Highest EPDS (depression) scores and the corresponding RBC folate levels and medication/supplement data for each *MTHFR C677T* genotype.**

| | | MTHFR genotype | | | |
| --- | --- | --- | --- | --- | --- |
| | Total | TT | CT | CC | p-value |
| | N = 305 | n = 33 | n = 116 | n = 156 | |
| **Highest postpartum EPDS score** | | | | | |
| Mean (SD) | 9.4 (5.2) | 10.4 (5.9) | 9.1 (5.1) | 9.4 (5.2) | 0.48 |
| **RBC folate (ng/ml)** | | | | | |
| Mean (SD) | 645.8 (218.5) | 722.0 (257.2) | 648.2 (217.1) | 627.8 (208.4) | 0.08 |
| **Taking a Folic Acid Supplement (n = 303)** | | | | | |
| no | 72 (23.8%) | 5 (15.6%) | 23 (19.8%) | 44 (28.3%) | 0.14 |
| yes* | 231 (76.2%) | 27 (84.4%) | 93 (80.2%) | 111 (71.6%) | |
| **Taking a daily Psychotropic Medication (n = 304)** | | | | | |
| no | 230 (75.7%) | 23 (69.7%) | 90 (78.3%) | 117 (75.0%) | 0.57 |
| yes | 74 (24.3%) | 10 (30.3%) | 25 (21.7%) | 39 (25.0%) | |

*n = 1 participant took 5-MTHF supplements (not folic acid).

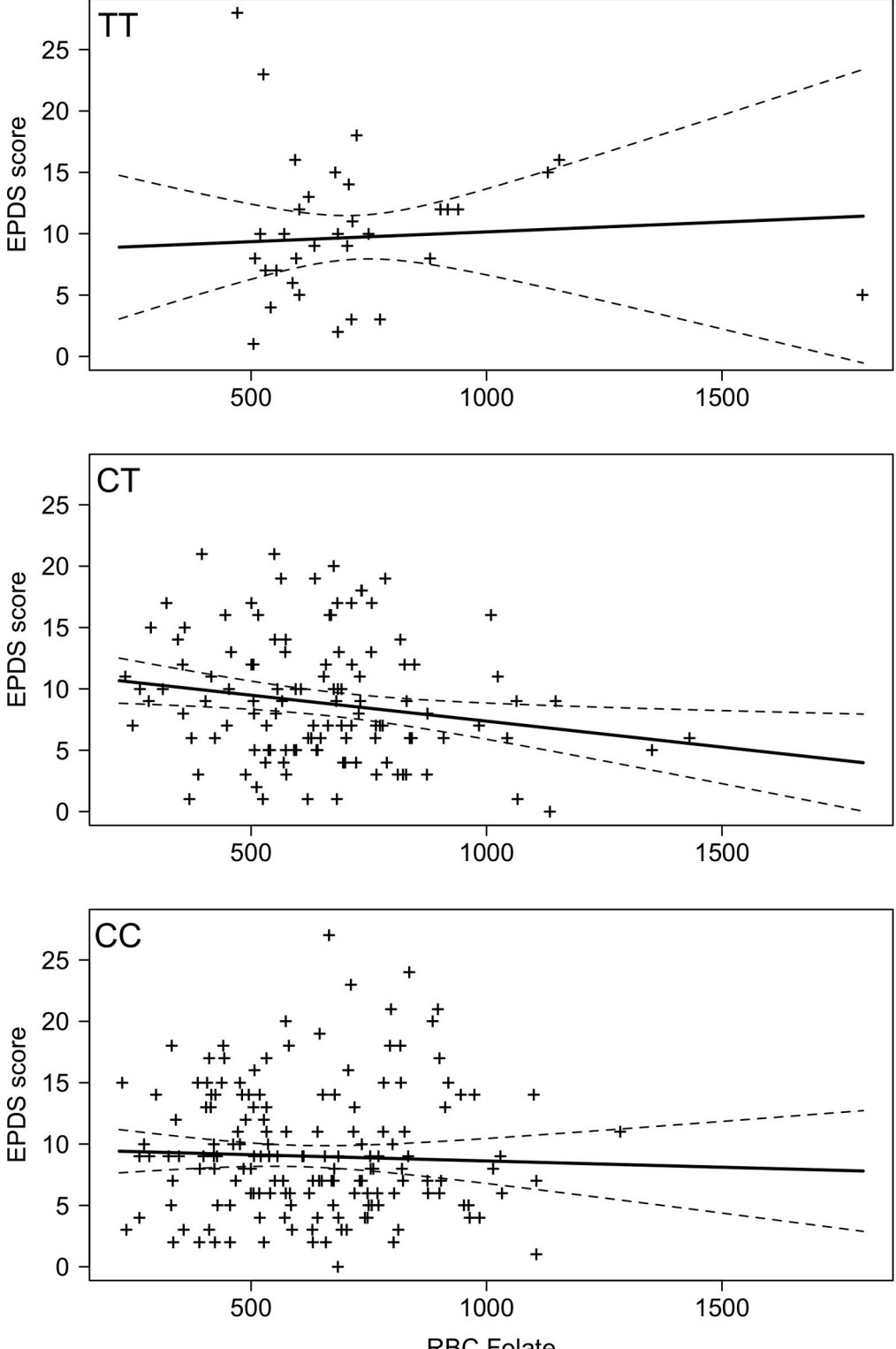

**Fig 1. (n = 305) EPDS (depression) score vs. RBC folate (ng/ml) score for each *MTHFR C677T (TT, CT, CC)* genotype (dashed line indicates confidence intervals).**

**Table 3. Highest CARS-M (mania) score and the corresponding RBC folate levels and medication/supplement data for each *MTHFR C677T* genotype.**

| | | MTHFR genotype | | | |
|---|---|---|---|---|---|
| | **Total** | **TT** | **CT** | **CC** | **p-value** |
| | **N = 233** | **n = 24** | **n = 90** | **n = 119** | |
| **Highest postpartum CARS-M score** | | | | | |
| Mean (SD) | 6.8 (3.8) | 7.46 (3.3) | 7.1 (3.7) | 6.5 (3.9) | 0.31 |
| **RBC folate (ng/ml)** | | | | | |
| Mean (SD) | 672.4 (227.9) | 818.9 (342.4) | 659.4 (199.1) | 652.7 (210.9) | 0.004 |
| **Taking a Folic Acid Supplement (n = 232)** | | | | | |
| no | 52(22.4%) | 4 (17.4%) | 13 (14.4%) | 35 (29.4%) | 0.03 |
| yes* | 180 (77.6%) | 19 (82.6%) | 77 (85.6%) | 84 (70.6%) | |
| **Taking a daily psychotropic medication** | | | | | |
| no | 176 (75.5%) | 16 (43.2%) | 73 (81.1%) | 87 (73.1%) | 0.22 |
| yes | 57 (24.5%) | 8 (33.3%) | 17 (18.9%) | 32 (26.9%) | |

* n = 1 participant took 5-MTHF supplements (not folic acid).

the first study to explore postpartum mania and any associations with *MTHFR* or folate levels. Our data suggest that, for women with a *MTHFR C677T* C allele (i.e. *CC* or *CT* genotypes), folate levels and mania symptoms may be inversely related, but that there is no relationship between folate levels and mania symptoms in women with a *TT* genotype. Interestingly, our data also showed that mean mania scores for women with a *TT* genotype did not significantly differ from those of women with *CC* or *CT* genotypes. While visually (see Fig 1) there is an appearance of an inverse relationship between depression scores and folate levels for women with CC and CT genotypes, that reflects the relationships found with mania, in this case, the relationship was not statistically significant. Given the small effect size of the difference in EPDS scores (when controlling for RBC folate levels) between *TT* genotypes and those with a *C* allele (*CC* and CT), it is possible that a relationship between *MTHFR* genotype and RBC folate levels (like we observed with mania symptoms) also exists for depressive symptoms, but the sample size was underpowered to detect it. Our data do not suggest a relationship between psychosis and folate levels and *MTHFR C677T* genotype.

*MTHFR C677T* was historically considered a strong candidate gene for a variety of psychiatric disorders (outside the perinatal setting) and was initially implicated in schizophrenia [45–50], bipolar and unipolar depression [50–53], though this association was not supported by genome wide association study (GWAS) findings [54, 55]. Our data suggest that *MTHFR C677T* genotype alone does not increase risk for postpartum psychopathology. Our findings are also broadly in line with previous studies that have shown that low folate (and thus subsequent high levels of homocysteine) is associated with various forms of psychopathology [56–59]. However, our findings suggest that this inverse relationship may only exist in the postpartum for women with a *MTHFR C677T C* allele, at least in terms of mania, and, perhaps, depressive symptoms.

One study that investigated the impact of folate levels on perinatal depression, without accounting for *MTHFR* genotype, did not find any relationship between plasma folate levels and depression in the *postpartum* period [24]. While our study also did not find a significant relationship of RBC folate with postpartum depression symptomatology, our results suggest that there may be an inverse relationship with other mood symptoms (mania), and possibly depression, when the effect of *MTHFR* genotype is considered.

While retrospective observational studies have suggested that self-reported perinatal folic acid supplementation can improve maternal depression symptomology several months

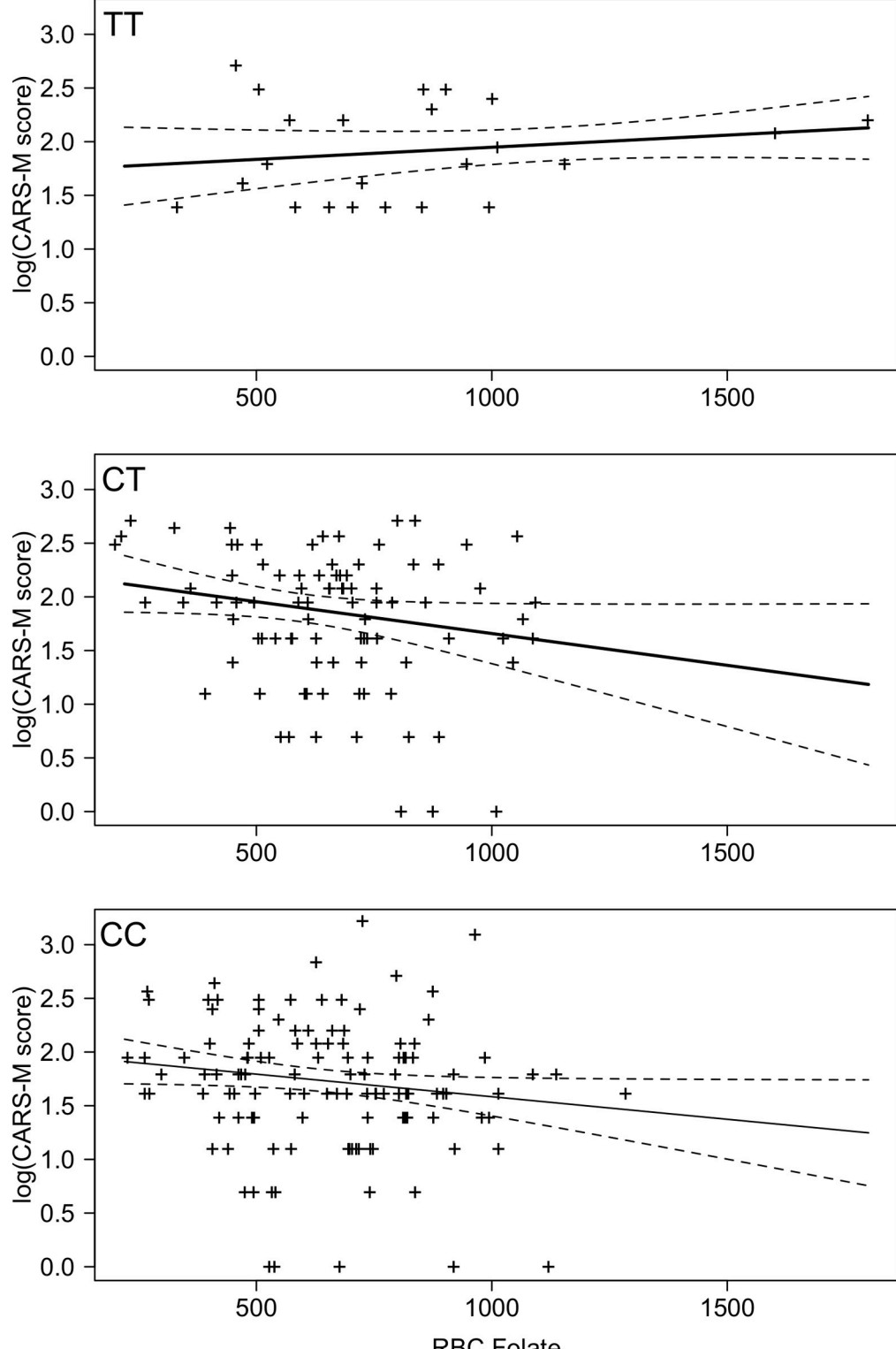

**Fig 2. (n = 233) CARS-M score vs. RBC folate (ng/ml) score for each *MTHFR C677T (TT, CT, CC)* genotype (dashed line indicates confidence intervals).**

**Table 4. Earliest presence of psychotic symptoms and the corresponding RBC folate levels and medication/supplement data for each *MTHFR C677T* genotype.**

| | | MTHFR genotype | | | |
|---|---|---|---|---|---|
| | **Total** | **TT** | **CT** | **CC** | **p-value** |
| | **N = 326** | **n = 37** | **n = 124** | **n = 165** | |
| **PANSS Psychosis Criteria Met** | | | | | |
| no | 261 (80.0%) | 33 (89.2%) | 96 (77.4%) | 132 (80.0%) | 0.31 |
| yes | 65 (20.1%) | 4 (10.8%) | 28 (22.6%) | 33 (20.0%) | |
| **RBC folate (ng/ml)** | | | | | |
| Mean (SD) | 705.0 (238.7) | 830.4 (369.8) | 698.9 (206.9) | 681.4(216.1) | 0.002 |
| **Taking a Folic Acid Supplement (n = 324)** | | | | | |
| no | 71 (21.9%) | 6 (16.7%) | 24 (19.4%) | 42 (25.6%) | 0.42 |
| yes* | 253 (78.1%) | 30 (83.3%) | 100 (80.6%) | 122 (73.9%) | |
| **Taking a daily psychotropic medication (n = 324)** | | | | | |
| no | 244 (75.3%) | 28 (75.7%) | 95 (77.2%) | 121 (73.8%) | 0.82 |
| yes | 80 (24.7%) | 9 (24.3%) | 28 (22.6%) | 43 (26.2%) | |

* n = 1 participant took 5-MTHF supplements (not folic acid).

postpartum, especially in women with a *MTHFR TT* genotype [28] our findings suggest that women with a *TT* genotype do not demonstrate an inverse relationship between physiological folate and mood symptoms (i.e. mania and perhaps, depression). Our results also suggest that overall there is no difference in physiological folate levels based on *MTHFR C677T* genotype (Table 1), but at times there may be higher physiological folate levels in those with *TT* genotypes (Tables 3 and 4), despite previous research suggesting that individuals who are *TT* homozygous would be more prone to folate deficiency [60]. These differences in our study may be due to the fact that a perinatal population is highly supplemented (e.g. in our study population the majority of women were taking 1 mg of folic acid daily, in addition to folate and folic acid consumed through diet) compared to studies in the general population. It is also possible that further studies of *MTHFR C677T* variants in pregnancy and postpartum cohorts are needed to fully understand the impact of pregnancy on enzyme function.

## Limitations

Since most of the women in our study were taking folic acid supplements, and all were living in Canada (a country with folic acid fortified foods), it is possible that an inverse relationship between physiological folate levels and psychiatric symptoms does exist for women who are *MTHFR TT* homozygotes; however, this relationship is not observable in populations with adequate folic acid supplementation (i.e. at a certain RBC folate level there may be a ceiling effect (for *TT* homozygotes) for the impact of folate on psychiatric symptoms). While there was no statistical difference in RBC folate levels (overall–see Table 1) between *CC*, *CT*, and *TT* genotypes, those with a *TT* genotype consistently had the highest RBC folate levels, which reached statistical significance when there was the highest levels of manic symptoms or presence/absence of psychotic symptoms, suggesting that the *TT* group was more than adequately supplemented, perhaps mitigating any negative impact of the *TT* genotype on psychopathology that would be observed when folic acid supplementation (and RBC folate) is less abundant. It is also possible that, while overall there was no difference in the proportions of women taking a folic acid supplements across the genotype groups (Table 1), participants with a *TT* genotype may, by chance, have been taking a greater amount of folic acid.

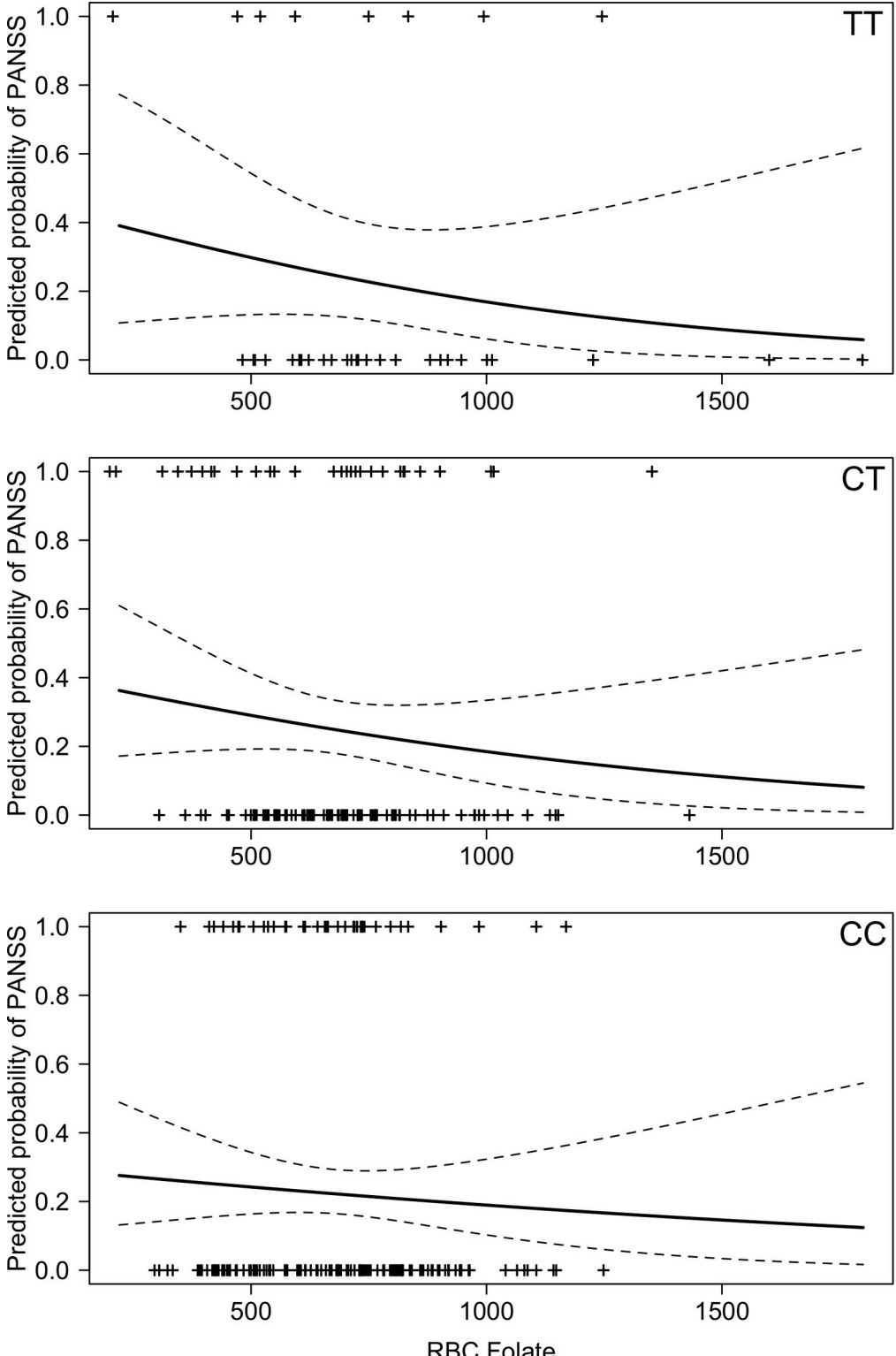

**Fig 3. (n = 326) Predicted probably of meeting threshold criteria for psychosis vs. RBC folate (ng/ml) for each *MTHFR C677T (TT, CT, CC)* genotype (dashed line indicates confidence intervals).**

Our data may be enriched for missing data at timepoints when women were the most depressed or experiencing psychotic symptoms (resulting in data only being available for a single timepoint, where symptoms may be less severe), as women may have cancelled study visits during periods of poorer mental health. This possibility increases the chance of a Type II error, whereby results indicate the absence of relationship that truly exists. Because our study population consisted of women who were already at a high risk for postpartum psychiatric disorders, due to their past history of a mood or psychotic disorder [2–10], our results may not be generalizable to other populations of postpartum women. As our study did not investigate the role of elevated homocysteine, future studies could explore relationships with *MTHFR C677T* genotype, psychopathology, and homocysteine levels.

## Conclusions

There may be a relationship between *MTHFR C677T* genotype, RBC folate levels, and risk for some forms of postpartum psychopathology, at least in the context of high-risk Canadian women with adequate folic acid supplementation. Further studies with larger samples of women may be needed to characterize the nuances in relationships between postpartum depression and mania symptoms in relation to *MTHFR C677T* genotype and folate levels.

## Supporting information

**S1 Table. Regression coefficients and data from linear and logistic regression analyses.** (DOCX)

## Acknowledgments

The authors thank all members of the Translational Psychiatric Genetics Group for their support, and the numerous volunteers over the years who helped with data entry. The authors would like to thank the participants who generously participated in the study.

## Author Contributions

**Conceptualization:** William Honer, Jehannine Austin.

**Data curation:** Emily Morris, Catriona Hippman, Arianne Albert, Caitlin Slomp, Angela Inglis, Prescilla Carrion, Rolan Batallones, Heather Andrighetti, Colin Ross, Roger Dyer.

**Formal analysis:** Emily Morris, Arianne Albert, Colin Ross.

**Funding acquisition:** William Honer, Jehannine Austin.

**Investigation:** Emily Morris, Catriona Hippman, Arianne Albert, Caitlin Slomp, Angela Inglis, Prescilla Carrion, Rolan Batallones, Heather Andrighetti, Colin Ross, Roger Dyer.

**Methodology:** Emily Morris, Catriona Hippman, Arianne Albert, Caitlin Slomp, Angela Inglis, Prescilla Carrion, Rolan Batallones, Heather Andrighetti, Colin Ross, Roger Dyer, Jehannine Austin.

**Project administration:** Emily Morris, Catriona Hippman, Caitlin Slomp, Angela Inglis, Prescilla Carrion, Rolan Batallones, Heather Andrighetti, Jehannine Austin.

**Resources:** Jehannine Austin.

**Supervision:** Colin Ross, Roger Dyer, Jehannine Austin.

**Visualization:** Arianne Albert.

**Writing – original draft:** Emily Morris.

**Writing – review & editing:** Catriona Hippman, Arianne Albert, Caitlin Slomp, Angela Inglis, Prescilla Carrion, Rolan Batallones, Heather Andrighetti, Colin Ross, Roger Dyer, William Honer, Jehannine Austin.

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
