## [Decision Letter · Decision Letter 0]

1 Sep 2020

PONE-D-20-16875

A prospective study to explore the relationship between MTHFRC677T genotype, physiological folate levels, and postpartum psychopathology in at-risk women

PLOS ONE

Dear Dr. Austin,

Thank you for submitting your manuscript to PLOS ONE. After careful consideration, we feel that it has merit but does not fully meet PLOS ONE’s publication criteria as it currently stands. Therefore, we invite you to submit a revised version of the manuscript that addresses the points raised during the review process.

The reviewers raised several important issues that should be resolved before considering publication in the Journal. Particularly, please pay attention to the methodological aspects.

We look forward to receiving your revised manuscript.

Kind regards,

Kyoung-Sae Na, M.D.

Academic Editor

PLOS ONE

Journal Requirements:

"This work was made possible through funding from the Canadian Institutes of

 Health Research (CIHR), JA was supported by BC Mental Health and Substance Use

Services, the Canada Research Chairs Program, the Michael Smith Foundation for Health

Research, and CIHR."

"JA received funding from the Canadian Institutes of Health Research (https://cihr-irsc.gc.ca/e/193.html) to conduct this work (CIHR MOP 82750). The funders had no role in study design, data collection and analysis, decision to publish, or preparation of the manuscript."

Reviewers' comments:

Reviewer's Responses to Questions

**Comments to the Author**

1. Is the manuscript technically sound, and do the data support the conclusions?

Reviewer #1: Partly

Reviewer #2: Yes

2. Has the statistical analysis been performed appropriately and rigorously? 

Reviewer #1: No

Reviewer #2: Yes

3. Have the authors made all data underlying the findings in their manuscript fully available?

Reviewer #1: No

Reviewer #2: Yes

4. Is the manuscript presented in an intelligible fashion and written in standard English?

Reviewer #1: Yes

Reviewer #2: Yes

5. Review Comments to the Author

Reviewer #1: The Authors analyse the relation between MTHFR C677T genotype, folate levels, and postpartum psychopathology in at-risk women. The Authors hypothesise that, in the first three months postpartum, compared to MTHFR CC homozygous women, TT homozygous women would have increased symptoms of postpartum depression and that this relationship would be moderated by physiological levels of red blood cell (RBC) folate. In addition, they conduct an exploratory analysis to assess the impact of MTHFR C677T genotypes and RBC folate levels on postpartum mania and postpartum psychosis.

The paper could benefit from clarification about data as well as the methodologies used. The Authors should provide more details on the modelling choices to help readers in having a clear understanding on the proposed study. Some important issues are listed below.

1. The Authors conducted a longitudinal observational study on a sample of 365 pregnant women with a history of a mood or psychotic disorder and that data collection occurred at 4 timepoints. However, the longitudinal aspect of this study is not exploited in the analysis where only the highest postpartum EPDS and CARS-M scores (from all the available postpartum timepoints) and the corresponding RBC folate levels were selected. Along the same line, RBC folate levels corresponding to the first postpartum timepoint for which the participant met criteria for psychosis, or to the earliest time point for participants which never met criteria for psychosis were chosen. All these choices might be due to a high presence of missing values at different time points but this is not clear from the text, where the only reason provided is linked to differences in timing for the emergence of symptoms of depression, mania and psychosis. The Authors should discuss in more detail why they do not perform a longitudinal statistical analysis.

2. To assess the hypothesised relationships, the Authors run a moderation analysis but they do not include any references to the literature on the moderation model. They should provide a clear and rigorous description of the moderation model implemented.

3. The Authors affirm that there are outliers in the data but they do not specify if the outliers affect only the dependent variables or also the folate levels, and neither do they provide statistical results to support their claim

4. To evaluate the relationships included in the moderation model, the Authors use robust linear regression. At line 192, they refer to “minimum maximum likelihood estimation”, which to the best of my knowledge does not exist. They are probably referring to the MM-estimator introduced by Yohai (1987). The Author should provide references for this method and specify the software used to perform the analysis.

Yohai, J. V. (1987). High Breakdown-Point and High Efficiency Robust Estimates for Regression, Annals of Statistics, 17, 4, 1662-1683.

5. How were post-hoc analyses conducted? Methodological aspect should be detailed in the manuscript.

6. I would suggest to move Table 1: Demographic information for each MTHFR C677T genotype, into the Supplementary data, and to include the results provided in Supplementary Table 1 in the manuscript, along with p-values, which are not reported in this Table.

7. In the Discussion, the Authors affirm that an inverse relationship between folate and depressive symptoms may exist in the postpartum for women with a MTHFR C677T 341 C. However, this conclusion is not supported by the analysis.

8. At lines 372-375, it reads “While there was no statistical difference in RBC folate levels […] between CC, CT, and TT genotypes,”. p-values reported in Tables 2 (p=0.08), 3 (p=0.004) and 4 (p=0.002) do not support this conclusion. This has to be clarified.

9. In the Limitation section, the Authors state that “participants with a TT genotype may, by chance, have been taking a greater amount of folic acid” (line 379). Could it be worth of investigation the inclusion of covariates, related to whether participants were taking a Folic Acid Supplement or a daily psychotropic medication, into the model?

Reviewer #2: The manuscript entitled “A prospective study to explore the relationship between MTHFRC677T genotype, physiological folate levels, and postpartum psychopathology in at-risk women” is an interesting and well structured study investigating the influence of MTHFR C677T polymorphism and folate blood levels on the onset of psychopathological disorders in the postpartum. The study design is methodologically flawless and statistical data analysis is appropriate and properly conducted. The graphs are immediately explanatory. It is also written in fluent and clear English. The topic is interesting for a wide audience. An increase in homocysteine levels resulting from folate deficiency and / or MTHFR mutations, has also been described as a possible risk factor for neuropsychiatric disorders, as well as for various other pathological situations, such as cardiovascular disease and osteoporosis, probably facilitating an increase in inflammatory factors, as suggested in a recent paper, to which the Authors could possibly refer (De Martinis M, Sirufo MM, Nocelli C, Fontanella L, Ginaldi L. Hyperhomocysteinemia is Associated with Inflammation, Bone Resorption, Vitamin B12 and Folate Deficiency and MTHFR C677T Polymorphism in Postmenopausal Women with Decreased Bone Mineral Density. Int J Environ Res Public Health 2020;17(12):4260.doi:10.3390/ijerph17124260). Are there data available on homocysteine levels in the women studied? The lack of this information should be considered among the limitations of the study. It might be interesting to evaluate them, maybe even in a future study. Since hyperomocysteinemia could be a pathogenetic mechanism that may help explain the effects of folate deficiency and MTHFR C677T polymorphism, I suggest that the Authors briefly address this topic in the discussion.

6. PLOS authors have the option to publish the peer review history of their article (what does this mean?). If published, this will include your full peer review and any attached files.

Reviewer #1: No

Reviewer #2: No

---

## [Author Response · Author response to Decision Letter 0]

7 Oct 2020

We thank the reviewers for the comments and feedback on our manuscript. We have documented our responses below (in bold italics) and have made the necessary changes to the manuscript.

Thank you. Changes have been made to reflect the formatting style.

"This work was made possible through funding from the Canadian Institutes of

 Health Research (CIHR), JA was supported by BC Mental Health and Substance Use

Services, the Canada Research Chairs Program, the Michael Smith Foundation for Health

Research, and CIHR."

"JA received funding from the Canadian Institutes of Health Research (https://cihr-irsc.gc.ca/e/193.html) to conduct this work (CIHR MOP 82750). The funders had no role in study design, data collection and analysis, decision to publish, or preparation of the manuscript."

Thank you for bringing this to our attention. We have deleted this text from our Acknowledgements. Please update our funding statement to say: 

This work was made possible through funding from the Canadian Institutes of

 Health Research (CIHR), JA was supported by BC Mental Health and Substance Use

Services, the Canada Research Chairs Program, the Michael Smith Foundation for Health

Research, and CIHR. JA received funding from the Canadian Institutes of Health Research (https://cihr-irsc.gc.ca/e/193.html) to conduct this work (CIHR MOP 82750). The funders had no role in study design, data collection and analysis, decision to publish, or preparation of the manuscript.

Our data contains potentially sensitive information. Further, in order to share de-identified data, our Research Ethics Board requires that participants explicitly consent. Unfortunately this was not included in our consent form for this study and we are unable to provide a de-identified data set to be made accessible. Data access requests can be sent to the University of British Columbia Clinical Research Ethics board (ethics.research.ubc.ca). 

Thank you for bringing this to our attention. We have included these captions in the manuscript and updated our in-text citations.

 5. Review Comments to the Author

Reviewer #1: The Authors analyse the relation between MTHFR C677T genotype, folate levels, and postpartum psychopathology in at-risk women. The Authors hypothesise that, in the first three months postpartum, compared to MTHFR CC homozygous women, TT homozygous women would have increased symptoms of postpartum depression and that this relationship would be moderated by physiological levels of red blood cell (RBC) folate. In addition, they conduct an exploratory analysis to assess the impact of MTHFR C677T genotypes and RBC folate levels on postpartum mania and postpartum psychosis.

The paper could benefit from clarification about data as well as the methodologies used. The Authors should provide more details on the modelling choices to help readers in having a clear understanding on the proposed study. Some important issues are listed below.

1. The Authors conducted a longitudinal observational study on a sample of 365 pregnant women with a history of a mood or psychotic disorder and that data collection occurred at 4 timepoints. However, the longitudinal aspect of this study is not exploited in the analysis where only the highest postpartum EPDS and CARS-M scores (from all the available postpartum timepoints) and the corresponding RBC folate levels were selected. Along the same line, RBC folate levels corresponding to the first postpartum timepoint for which the participant met criteria for psychosis, or to the earliest time point for participants which never met criteria for psychosis were chosen. All these choices might be due to a high presence of missing values at different time points but this is not clear from the text, where the only reason provided is linked to differences in timing for the emergence of symptoms of depression, mania and psychosis. The Authors should discuss in more detail why they do not perform a longitudinal statistical analysis.

 We appreciate this comment and the opportunity to provide further clarity in the manuscript. Indeed, as the reviewer mentions, there are some missing values, resulting in not all participants having data at each timepoint, limiting the number of participants who have data across all timepoints. The focus of our study; however, was not on the longitudinal evolution of psychiatric symptoms but, rather to observe the most severe symptoms (depression and mania) or if they every met criteria (psychosis), in our study time frame to test differences between genotypes. We have added the limitation of missing data to our limitation section to reflect how this may have influenced our findings.

2. To assess the hypothesised relationships, the Authors run a moderation analysis but they do not include any references to the literature on the moderation model. They should provide a clear and rigorous description of the moderation model implemented.

We have modified the analysis section of our manuscript to provide more clarity on our model and have included references where necessary. 

3. The Authors affirm that there are outliers in the data but they do not specify if the outliers affect only the dependent variables or also the folate levels, and neither do they provide statistical results to support their claim

We thank the reviewer for highlighting this – we have added further details to our analysis section to better explain our analyses of outliers; specifically that their presence was assessed using diagnostic regression plots of non-linear models. 

4. To evaluate the relationships included in the moderation model, the Authors use robust linear regression. At line 192, they refer to “minimum maximum likelihood estimation”, which to the best of my knowledge does not exist. They are probably referring to the MM-estimator introduced by Yohai (1987). The Author should provide references for this method and specify the software used to perform the analysis.

Yohai, J. V. (1987). High Breakdown-Point and High Efficiency Robust Estimates for Regression, Annals of Statistics, 17, 4, 1662-1683.

The reviewer is correct and we have revised the manuscript and have have added appropriate references regarding our use of MM-estimation to mitigate the effects of outliers and highly influential points.

5. How were post-hoc analyses conducted? Methodological aspect should be detailed in the manuscript.

We have provided more specific details about our “post-hoc” analysis in the manuscript.

6. I would suggest to move Table 1: Demographic information for each MTHFR C677T genotype, into the Supplementary data, and to include the results provided in Supplementary Table 1 in the manuscript, along with p-values, which are not reported in this Table.

We thank the reviewers for this suggestion. The p-values associated with the Supplementary data appear in the main body of the manuscript. As table 1 contains information fundamental to interpretation of the data, we have left this in the main body of the manuscript. 

7. In the Discussion, the Authors affirm that an inverse relationship between folate and depressive symptoms may exist in the postpartum for women with a MTHFR C677T 341 C. However, this conclusion is not supported by the analysis.

We appreciate the opportunity to clarify this for the reviewer. The reviewer is correct, there is no relationship. We have edited our previously ambiguous wording to make this clearer in the manuscript.

8. At lines 372-375, it reads “While there was no statistical difference in RBC folate levels […] between CC, CT, and TT genotypes,”. p-values reported in Tables 2 (p=0.08), 3 (p=0.004) and 4 (p=0.002) do not support this conclusion. This has to be clarified.

We thank the reviewer for highlighting this discrepancy. This statement in the limitations section was an oversight on our part. We have edited it to more accurately reflect our data.

9. In the Limitation section, the Authors state that “participants with a TT genotype may, by chance, have been taking a greater amount of folic acid” (line 379). Could it be worth of investigation the inclusion of covariates, related to whether participants were taking a Folic Acid Supplement or a daily psychotropic medication, into the model?

We thank the reviewer for this comment. We have included whether participants were taking a folic acid supplement, and psychotropic medication in Table 1 and compared against genotype groups, for which there were no significant differences - with the majority of individuals taking a folic acid supplement, and a minority taking medications for all genotype groups. 

Reviewer #2: The manuscript entitled “A prospective study to explore the relationship between MTHFRC677T genotype, physiological folate levels, and postpartum psychopathology in at-risk women” is an interesting and well structured study investigating the influence of MTHFR C677T polymorphism and folate blood levels on the onset of psychopathological disorders in the postpartum. The study design is methodologically flawless and statistical data analysis is appropriate and properly conducted. The graphs are immediately explanatory. It is also written in fluent and clear English. The topic is interesting for a wide audience.

 An increase in homocysteine levels resulting from folate deficiency and / or MTHFR mutations, has also been described as a possible risk factor for neuropsychiatric disorders, as well as for various other pathological situations, such as cardiovascular disease and osteoporosis, probably facilitating an increase in inflammatory factors, as suggested in a recent paper, to which the Authors could possibly refer (De Martinis M, Sirufo MM, Nocelli C, Fontanella L, Ginaldi L. Hyperhomocysteinemia is Associated with Inflammation, Bone Resorption, Vitamin B12 and Folate Deficiency and MTHFR C677T Polymorphism in Postmenopausal Women with Decreased Bone Mineral Density. Int J Environ Res Public Health 2020;17(12):4260.doi:10.3390/ijerph17124260). Are there data available on homocysteine levels in the women studied? The lack of this information should be considered among the limitations of the study. It might be interesting to evaluate them, maybe even in a future study. Since hyperomocysteinemia could be a pathogenetic mechanism that may help explain the effects of folate deficiency and MTHFR C677T polymorphism, I suggest that the Authors briefly address this topic in the discussion.

We thank the reviewer for this suggestion and have edited the limitations section accordingly and will explore the possibilities to examine this in our own future work.

---

## [Decision Letter · Decision Letter 1]

1 Dec 2020

A prospective study to explore the relationship between MTHFRC677T genotype, physiological folate levels, and postpartum psychopathology in at-risk women

PONE-D-20-16875R1

Dear Dr. Austin,

We’re pleased to inform you that your manuscript has been judged scientifically suitable for publication and will be formally accepted for publication once it meets all outstanding technical requirements.

Kind regards,

Kyoung-Sae Na, M.D.

Academic Editor

PLOS ONE

Additional Editor Comments (optional):

Reviewers' comments:

Reviewer's Responses to Questions

**Comments to the Author**

1. If the authors have adequately addressed your comments raised in a previous round of review and you feel that this manuscript is now acceptable for publication, you may indicate that here to bypass the “Comments to the Author” section, enter your conflict of interest statement in the “Confidential to Editor” section, and submit your "Accept" recommendation.

Reviewer #1: All comments have been addressed

Reviewer #2: All comments have been addressed

2. Is the manuscript technically sound, and do the data support the conclusions?

Reviewer #1: Yes

Reviewer #2: Yes

3. Has the statistical analysis been performed appropriately and rigorously? 

Reviewer #1: Yes

Reviewer #2: Yes

4. Have the authors made all data underlying the findings in their manuscript fully available?

Reviewer #1: Yes

Reviewer #2: Yes

5. Is the manuscript presented in an intelligible fashion and written in standard English?

Reviewer #1: Yes

Reviewer #2: Yes

6. Review Comments to the Author

Reviewer #1: (No Response)

Reviewer #2: The paper is acceptable for publication, although some interesting topics that we suggested to address in this first revision have been missed.

7. PLOS authors have the option to publish the peer review history of their article (what does this mean?). If published, this will include your full peer review and any attached files.

Reviewer #1: No

Reviewer #2: No

---

## [Editor Report · Acceptance letter]

3 Dec 2020

PONE-D-20-16875R1 

A prospective study to explore the relationship between *MTHFR C677T* genotype, physiological folate levels, and postpartum psychopathology in at-risk women 

Dear Dr. Austin:

I'm pleased to inform you that your manuscript has been deemed suitable for publication in PLOS ONE. Congratulations! Your manuscript is now with our production department. 

Kind regards, 

on behalf of

Dr. Kyoung-Sae Na 

Academic Editor

PLOS ONE